# Suppression of B-Cell Activation by Human Cord Blood-Derived Stem Cells (CB-SCs) through the Galectin-9-Dependent Mechanism

**DOI:** 10.3390/ijms25031830

**Published:** 2024-02-02

**Authors:** Wei Hu, Xiang Song, Haibo Yu, Sophia Fan, Andrew Shi, Jingyu Sun, Hongjun Wang, Laura Zhao, Yong Zhao

**Affiliations:** 1Center for Discovery and Innovation, Hackensack Meridian Health, Nutley, NJ 07110, USA; wei.hu@hmh-cdi.org (W.H.);; 2Throne Biotechnologies, Paramus, NJ 07652, USA; 3Department of Chemistry and Chemical Biology, Stevens Institute of Technology, Hoboken, NJ 07030, USA; jsun20@stevens.edu (J.S.); hwang2@stevens.edu (H.W.)

**Keywords:** cord blood-derived stem cells, Stem Cell Educator therapy, B cells, galectin-9, immune modulation, type 1 diabetes, autoimmune diseases

## Abstract

We developed the Stem Cell Educator therapy among multiple clinical trials based on the immune modulations of multipotent cord blood-derived stem cells (CB-SCs) on different compartments of immune cells, such as T cells and monocytes/macrophages, in type 1 diabetes and other autoimmune diseases. However, the effects of CB-SCs on the B cells remained unclear. To better understand the molecular mechanisms underlying the immune education of CB-SCs, we explored the modulations of CB-SCs on human B cells. CB-SCs were isolated from human cord blood units and confirmed by flow cytometry with different markers for their purity. B cells were purified by using anti-CD19 immunomagnetic beads from human peripheral blood mononuclear cells (PBMCs). Next, the activated B cells were treated in the presence or absence of coculture with CB-SCs for 7 days before undergoing flow cytometry analysis of phenotypic changes with different markers. Reverse transcription-polymerase chain reaction (RT-PCR) was utilized to evaluate the levels of galectin expressions on CB-SCs with or without treatment of activated B cells in order to find the key galectin that was contributing to the B-cell modulation. Flow cytometry demonstrated that the proliferation of activated B cells was markedly suppressed in the presence of CB-SCs, leading to the downregulation of immunoglobulin production from the activated B cells. Phenotypic analysis revealed that treatment with CB-SCs increased the percentage of IgD^+^CD27^−^ naïve B cells, but decreased the percentage of IgD^−^CD27^+^ switched B cells. The transwell assay showed that the immune suppression of CB-SCs on B cells was dependent on the galectin-9 molecule, as confirmed by the blocking experiment with the anti-galectin-9 monoclonal antibody. Mechanistic studies demonstrated that both calcium levels of cytoplasm and mitochondria were downregulated after the treatment with CB-SCs, causing the decline in mitochondrial membrane potential in the activated B cells. Western blot exhibited that the levels of phosphorylated Akt and Erk1/2 signaling proteins in the activated B cells were also markedly reduced in the presence of CB-SCs. CB-SCs displayed multiple immune modulations on B cells through the galectin-9-mediated mechanism and calcium flux/Akt/Erk1/2 signaling pathways. The data advance our current understanding of the molecular mechanisms underlying the Stem Cell Educator therapy to treat autoimmune diseases in clinics.

## 1. Introduction

Human cord blood-derived stem cells (CB-SCs) display a unique phenotype, with both embryonic and hematopoietic markers that distinguish them from other known types of stem cells, including hematopoietic stem cells (HSCs) and mesenchymal stem cells (MSCs). CB-SCs display the leukocyte common antigen CD45 and embryonic stem cell markers such as the transcription factor OCT 3/4 and SOX2, as well as the immune modulation-associated markers CD270 and CD274, but are negative for hematopoietic stem cell (HSC) marker CD34 and mesenchymal stem cell (MSC) markers CD90 and CD105 [1]. Our previous studies demonstrated that human CB-SCs display multiple immune modulations on T cells and monocytes/macrophages via surface molecules and released exosomes [1,2]. Based on CB-SCs’ immunomodulation, we developed the Stem Cell Educator® (SCE) therapy to treat immune dysfunction-associated diseases, including type 1 diabetes (T1D), type 2 diabetes (T2D), and alopecia areata (AA) [3,4,5], through international multicenter clinical trials in the United States, China, and Spain. SCE therapy circulates a patient’s peripheral blood mononuclear cells (PBMCs) through a blood cell separator, cocultures their immune cells with adherent CB-SCs in vitro, and then returns the “educated” immune cells back to the patient’s blood circulation. Our clinical data have successfully demonstrated the safety and clinical efficacy of SCE therapy in reversing the autoimmunity, promoting the regeneration of islet β cells, and improving the metabolic control in T1D and T2D patients [3,4].

B cells have an important role in maintaining homeostasis and the adaptive immune response through antibody production, antigen presentation, and the production of multiple cytokines [6,7]. Dysfunctions of B cells, including the production of auto-antibodies and the loss of regulatory B-cell function, are actively contributed to by the pathogenesis of diabetes [8,9,10,11,12,13] and multiple autoimmune diseases [14,15]. For example, their roles as antibody-producing cells in systemic lupus erythematosus (SLE) [16] and antigen-presenting cells in T1D and rheumatoid arthritis (RA) have been well recognized [8,17]. Therefore, it is essential to correct B cell-associated immune dysfunctions for the treatment of autoimmune diseases.

The galectin family comprises the glycan-binding proteins that are expressed by diverse types of cells and tissues, including immune and non-immune cells [18,19]. Galectins typically recognize the glycoproteins/glycolipids through the conserved carbohydrate recognition domains (CRDs) and bind primarily to β-galactoside carbohydrates with high specificity [20], which is present on a variety of glycoproteins on the surface of nearly every cell [18]. In the immune system, galectins are important regulators in the innate and adaptive immune responses by regulating a variety of immune cell activations, maturations, and other activities. Galectin-1, -3, and -9 have shown different effects on the functioning of T cells by modulating their development, activation, and differentiation [21,22,23]. However, the actions of galectins in B cells have only recently begun to be deciphered. Emerging evidence demonstrated that galectins play important roles in the signaling transduction and modulations of B-cell development, differentiation, activation, and antibody productions [24]. Specifically, galectin-9 (Gal-9) is a 34–39 kDa tandem-repeat-type protein, consisting of two carbohydrate recognition domains, which is found in immune cells, endothelial cells, and stem cells [25,26]. Gal-9 could not only suppress T-cell activation via the Tim-3 or PD-1 receptor on T cells [18,27], but could also suppress B-cell activation through the B-cell receptor [28,29]. Giovannone et al. reported that the common leukocyte antigen CD45 is the major Gal-9 receptor in human B cells. The binding of CD45 by Gal-9 suppresses calcium signaling via a Lyn-CD22-SHP-1 dependent mechanism and blunts B-cell activation [28]. To date, our mechanistic studies have confirmed the immune modulations of SCE therapy on the activated T cells, autoimmune memory T cells [30], regulatory T cells (Tregs) [3], and monocytes/macrophages [2]. The effects of CB-SCs on B cells remained elusive. Here, we demonstrated the direct immune modulation of CB-SCs on the activated B cells via the Gal-9-mediated mechanism, leading to the marked suppression of B-cell proliferation and phenotypic changes.

## 2. Results

### 2.1. CB-SCs Suppressed the Proliferation of Activated B Cells

Initially, the purity of CB-SCs was characterized by flow cytometry with CB-SC-associated markers including leukocyte common antigen CD45, embryonic stem (ES) cell markers OCT3/4 and SOX2, hematopoietic stem cell marker CD34, and immune tolerance-related markers CD270 and CD274. The findings revealed that CB-SCs highly expressed CD45, OCT3/4, SOX2, CD270, and CD274, but did not express CD34 (Figure 1A). Therefore, CD45 and Oct3/4 were specially selected and routinely utilized for the purity confirmation of CB-SCs (at ≥95% of CD45^+^ OCT3/4^+^ CB-SC).

To explore the immune modulation effects of CB-SCs on B cells, CB-SCs were cocultured with B cells at different ratios of CB-SCs. B cells at different ratios (e.g., 1:2, 1:5, and 1:10) and B cells only were activated by the cocktails (anti-IgM, rCD40L, IL-2, IL-4, IL-10, IL-21) [28]. The purity of positively selected CD19^+^ cells was more than 95%, as determined by flow cytometry with Korman orange-conjugated mouse antihuman CD19 antibody. The proliferation of B cells was examined by flow cytometry, after carboxyfluorescein succinimidyl (CFSE) staining and in combination with the propidium iodide staining, to determine the dead cells. The results revealed that there were no differences in the percentages of dead cells among different groups of CB-SC treatments relative to that of CB-SC-untreated B cells (Appendix A). However, the percentages of B-cell proliferation markedly declined from 91.52 ± 5.31 to 65.84 ± 8.24 at the ratio of 1:2 (*p* < 0.005) and 76.78 ± 7.44 at the ratio of 1:5 (*p* < 0.05), respectively (Figure 1B,C). The data indicated the suppression of CB-SCs on the B-cell proliferation.

### 2.2. CB-SCs Inhibited Immunoglobulin Production

Immunoglobulins are proteins that are secreted by plasma B cells and are present on the surface of B cells (e.g., IgD). They are assembled from identical couples of heavy and light chains. Based on the difference among heavy chains, immunoglobulins are characterized by five classes of Ig: IgM, IgG, IgA, IgE, and IgD. To explore the effects of CB-SCs on B cells, Ig productions were examined by using flow cytometry. Comparing with the Ig productions of CB-SC-untreated B cells, the data showed that CB-SCs could markedly inhibit Ig production at a ratio of CB-SC/B-cells of 1:5. As shown in Figure 2, a significant reduction in several Ig classes was detected, including IgG1 production (*p* = 0.0068) (Figure 2A), IgG2 (*p* = 0.330) (Figure 2B), IgG3 (*p* = 0.002) (Figure 2C), IgG4 (*p* = 0.0035) (Figure 2D), IgA (*p* = 0.048) (Figure 2E), and IgM (*p* = 0.019) (Figure 2F).

### 2.3. Modulation of CB-SC on Naïve and Memory B Cells

CD27 and IgD are the widely accepted biomarkers that are used to characterize B cells into memory and naïve subsets, such as naïve B cells (CD27^−^IgD^+^), switched memory B cells (CD27^+^IgD^−^), and non-switched memory B cells (CD27^+^ IgD^+^) [31]. To further explore the effects of CB-SCs on memory B cells, the activated B cells were treated with or without CB-SCs. Flow cytometry established that both the percentages of naïve B cells and switched B cells were significantly downregulated after the B-cell activation. However, their percentages were markedly changed after the treatment with CB-SCs (Figure 3A,B,D). The percentage of naïve B cells was increased from 8.79 ± 2.14 for CB-SC-untreated B cells to 28.23 ± 4.82 for CB-SC-treated B cells at a CB-SC/B cell ratio of 1:2. (*p* = 0.0002, Figure 3B). On the other hand, the percentage of switched memory B cells decreased from 63.53 ± 6.85 for CB-SC-untreated B cells to 48.75 ± 3.09 for CB-SC-treated B cells at a CB-SC/B cell ratio of 1:2. (*p* = 0.0004, Figure 3D). Notably, the percentage of non-switched CD27^+^IgD^+^ memory B cells failed to mark changes before and after the treatment with CB-SCs (Figure 3C). The data suggest the modulation effect of CB-SCs on the B-cell differentiation.

### 2.4. CB-SC-Mediated B Cell Suppression via the Cell–Cell Contact Regulation

Our previous studies demonstrated that the CB-SC-mediated immune modulations of T cells were achieved through multiple mechanisms, such as PD-L1/PD1-mediated cell–cell inhibition and releasing soluble factors (e.g., nitric oxide and transforming growth factor-β1) [1]. To elaborate on whether the cell–cell contact or soluble factors were involved in the immune modulations of CB-SCs on B cells, transwell-based coculture experiments were accordingly performed, in which CB-SCs and stimulated B cells were either in direct contact with or physically separated from the transwell inserts. Flow cytometry revealed that the suppression of CB-SCs on B-cell proliferation was abolished in the transwell coculture system (Figure 4A,B), highlighting the possible involvement of the surface or intracellular molecules that are expressed on CB-SCs, which contributes to the suppression of B cells.

### 2.5. The Expression of Gal-9 on CB-SCs Acts as a Key Molecule Contributing to the B-Cell Modulation

Our previous works demonstrated that CB-SCs displayed PD-L1 (CD274) [32] and herpesvirus entry mediator (HVEM, CD270) [5], which contributed to the immune modulations of CB-SCs on T cells. To reveal the molecular mechanisms underlying the immune modulation of CB-SCs on B cells, we further explored other potential surface molecules such as galectins [28,33], which are involved in the immune suppression of human mesenchymal stem cells (MSC) on both T and B cells [26,34]. Galectins have been recognized as essential regulators contributing to the induction of an immune tolerance and homeostasis, therefore functioning as attractive therapeutic targets for attenuating autoimmune and inflammation disorders [23,35]. To determine whether galectins were involved in the immune modulation of CB-SCs on B cells, we first analyzed the profile of galectin mRNA expressions including Gal-1, -2, -3, -4, -7, -8, -9, -10, -12, -13, and -14 (Figure 5A, Appendix A). The RT-PCR results showed that there were higher expressions of Gal-1, -2, -3, -4, -7, -8, and -9 mRNA than those of others (e.g., Gal-10, -12, -13, and -14) in CB-SCs (Appendix A). In comparison with the changes in other galectin mRNA levels, the level of Gal-9 mRNA expression was markedly increased to approximately 10-fold in the CB-SCs that were cocultured with the activated B cells for 48 h (Figure 5A). To confirm whether the protein level of Gal-9 was also upregulated, we performed flow cytometry with Gal-9 mAb. The flow data revealed that the median fluorescence intensity (MFI) in CB-SCs was significantly increased in the presence of activated B cells (Figure 5B,C). To further prove the involvement of CB-SC-expressed Gal-9 in the immune modulation of CB-SCs on B cells, we performed a blocking experiment by neutralizing Gal-9 mAb. It was found that the neutralization of Gal-9 with Gal-9 mAb marginally reversed the CB-SC-induced proliferation suppression of activated B cells compared to that without a Gal-9 mAb blocking group (*p* = 0.025, Figure 5D). This finding verified that the CB-SC-expressed Gal-9 contributed to the immune modulation of CB-SCs on activated B cells.

### 2.6. Gal-9-Dependent Suppression of Calcium Flux in B Cells by CB-SCs

The activation and proliferation of B cells are initiated by the B cell receptor (BCR), which triggers a number of signaling cascades [36]. Increases in the intracellular Ca^2+^ levels are essential in order to tune the B-cell responses and subsequent development post the BCR activation [37]. To further explore the molecular mechanism underlying the inhibition of B-cell proliferation by the treatment with CB-SCs, we examined the changes in cytosolic and mitochondrial Ca^2+^ levels in the CB-SC-treated B cells by flow cytometry after being stimulated with B cell-dependent activation cocktails. The cytosolic calcium level was assessed by using the Fluor-4 staining. As shown in Figure 6A, the median fluorescence intensity of Fluo-4^+^-activated B cells was markedly downregulated in the presence of CB-SC at a CB-SC/B cell ratio of 1:5 (*p* = 0.004). The suppressive effect on the cytosolic Ca^2+^ levels was reversed after the blocking with Gal-9 mAb (Figure 6A). The direct influence of Gal-9 on the cytosolic Ca^2+^ level was further evidenced by the attenuated cytosolic Ca^2+^ level upon 0.5 μg/mL recombinant Gal-9 (Figure 6A). Using the Rhod-2 staining as an indicator for the mitochondrial calcium, we were also able to demonstrate that the mitochondrial Ca^2+^ levels in the stimulated B cells were significantly reduced after the treatment with CB-SCs at a ratio of 1:5 of CB-SC to B cells (Figure 6B). Similar to the changes in the cytosolic Ca^2+^ levels, the mitochondrial Ca^2+^ levels in the CB-SC-treated B cells could be rebounded back upon blocking with the Gal-9 mAb (Figure 6B). In addition, a reduction in the mitochondrial membrane potential was also noticed with the stimulated B cells after the treatment with CB-SCs, and such a reduction was almost completely recovered after the blocking treatment with Gal-9 mAb (Figure 6C and Appendix A). All these data suggest that the Gal-9-mediated Ca^2+^ signaling pathway plays a key role in the CB-SC-induced modulation of the activated B cells.

Additionally, Western blotting was also performed to further investigate the BCR downstream molecules in order to confirm the direct participation of BCR in the Gal-9-mediated regulation. As shown in Appendix A, both the BCR downstream molecules phospho-Akt and phospho-Erk1/2 were drastically upregulated in the stimulated B cells despite the comparable total protein levels. Interestingly, the treatment with CB-SCs or recombinant human Gal-9 (0.5 µg/mL) significantly reduced the p-Akt and p-Erk1/2 levels to those of the unstimulated controls. Such inhibitory effects of CB-SCs on the p-Akt and p-Erk1/2 were almost completely abolished after blocking with Gal-9 mAb (Appendix A). The obtained data strongly indicate that Gal-9 expressed on CB-SCs contributed to the immune modulation of CB-SCs on B cells via the regulation of the Ca^2+^ flux and the phosphorylation of the Akt and Erk1/2 signaling pathways.

## 3. Discussion

Over the last 12 years, CB-SCs have been utilized in international multicenter clinical trials and designated to Stem Cell Educator^®^ therapy for the treatment of autoimmune diseases, including type 1 diabetes [3,30], alopecia areata [5], and other chronic metabolic inflammation-associated diseases (e.g., type 2 diabetes [4]). Mechanistic studies have demonstrated that CB-SCs displayed strong immune modulation on T cells and monocytes such as the inhibition of T-cell activation and proliferation, percentage reductions of effector memory T cells (T_EM_) [30], and the induction of the differentiation of monocytes into anti-inflammation type 2 macrophages (M2) [2]. However, the mechanistic modulation of CB-SC on B cells is yet to unfold. Here, we demonstrated the CB-SC-induced immunomodulation on activated B cells by inhibiting the B-cell proliferation and the activation of naïve B cells, downregulating the differentiation of switched memory B cells, and reducing the production of immunoglobulins. These findings clearly advance our understanding of the molecular mechanism of Stem Cell Educator therapy for the treatment of T1D and other autoimmune diseases.

B cells are important effector cells that are involved in the pathogenesis of autoimmune diseases through the production of auto-antibodies, the promotion of CD4^+^ T cell responses via antigen presentation, and the release of inflammatory cytokines (e.g., TNF-α and IL-6) [15,38]. Increasing evidence indicates the importance of B cell-mediated autoimmunity in the pathogenesis of T1D, even though T cells are generally considered the major pathogenic effector cells contributing to the destruction of islet β cells. Researchers found that blocking B cells or impairing the B cell function will significantly decrease the incidence of diabetes in non-obese diabetic mice [39,40]. Additionally, the depletion of B cells with antihuman CD20 antibody (Rituximab) markedly preserved the islet β-cell function and improved C-peptide levels after 1 year of follow-up in recent-onset T1D patients [41]. The current study demonstrated that CB-SCs markedly suppressed the proliferation of activated B cells and reduced the antibody productions in these activated B cells. Therefore, these data suggest the clinical translational potential of Stem Cell Educator therapy to treat other B cell-mediated autoimmune diseases. These therapeutic effects on B cells will be tested in our ongoing FDA-approved clinical trial (IND #19247, ClinicalTrials.gov ID: NCT04011020) by using Stem Cell Educator therapy to treat T1D subjects.

To date, the characterization of the B-cell phenotype with multiparameter flow cytometry has identified several B-cell subpopulations, including CD27^+^IgD^+^ non-switched memory B cells and CD27^+^IgD^−^ switched memory B cells, which may represent a biomarker for some autoimmune diseases. For instance, the percentage of switched memory B cells increased in systemic lupus erythematosus and rheumatoid arthritis may contribute to this [42,43,44]. Our flow cytometry analysis substantiated that the percentage of switched memory B cells was markedly reduced after the treatment with CB-SCs in a dose-dependent manner. Notably, the percentage of naïve B cells (CD27^−^IgD^+^) was increased, highlighting the modulation of CB-SCs on B-cell differentiation with the reduction in memory B cells.

To elucidate the molecular basis of SCE therapy, previous studies have identified several molecular and cellular pathways that alter autoimmune T cells and the functions of pathogenic monocytes/macrophages (Mo/Mϕs) to elicit immune tolerance via (1) the expression of the autoimmune regulator (AIRE) in CB-SCs, which is a master transcriptional regulator that acts to eliminate the self-antigen-reactive T cells in the thymus and is controlled by the activation of the receptor activator of NF-κB signaling pathway [45]; (2) the secretion of CB-SC-derived exosomes (cbExosomes), which polarize human blood Mo/Mϕ into type 2 macrophages (M2) [2], further contributing to immune tolerance and preventing β-cell destruction; and (3) the migration of platelet-derived mitochondria (pMitochondria) to islets, which are absorbed by pancreatic islets and contribute to an improved proliferation of human islet β cells [46].

Our current studies revealed the direct immune modulation of CB-SCs on activated B cells through the Gal-9, as demonstrated by transwell coculture and a blocking experiment with anti-Gal-9 mAb. What is more, further mechanistic studies confirmed that Gal-9 expressed on CB-SCs directly contributed to the regulation of the Ca^2+^ flux and phosphorylation of Akt and Erk1/2 signaling pathways in the stimulated B cells. Waters and colleagues reported an increase in the oxidative phosphorylation and mitochondrial membrane potential (Δψm) among the stimulated B cells [47], which was consistent with our current data showing the enhanced median fluorescence intensity of TMRE staining. Notably, the Δψm of stimulated B cells was substantially reduced in the Gal-9-dependent manner.

Galectins are β-galacotoside-binding lectins, which can be expressed by different types of stem cells and act as regulators of immune cell function [48], especially galectin-3 and Gal-9. Galectin-3 suppresses the activation of TCR-mediated signal transduction, while Gal-9 binds T cell Ig mucin-3 (Tim-3) and induces negative regulate T helper 1 immunity [49]. Our current data confirmed that galectin-1, 2, 3, 4, 7, 8, and 9 were highly expressed in CB-SCs. Gal-9 contributed to the immune modulation of CB-SCs on activated B cells. In line with our current study, the expression of Gal-9 on mesenchymal stem cells displays immune modulations on T/B cells and a therapeutic potential in experimental endotoxemia [26,50]. Interestingly, both the mRNA and protein levels of Gal-9 expression were markedly increased in CB-SCs after coculture with the activated B cells. The detailed molecular mechanism needs to be explored. Gal-9 was not only located in the cytoplasm and on the cellular membrane, but also acted as a soluble factor that is involved in the immune modulation [18,27]. To test this possibility, we found that CB-SC-released Gal-9 was less than 10% of the total CB-SC-derived Gal-9 after 3 days of culture (Appendix A). Therefore, Gal-9 produced by CB-SCs (including both intracellular and cell surface Gal-9) displayed more potential than the soluble form of CB-SC-secreted Gal-9 during the B-cell immune modulation. Further studies are needed to discriminate the roles of intracellular and cell surface Gal-9 expressed by CB-SCs during their B-cell modulations. Giovannone et al. reported that Gal-9 can directly bind to the poly-LacNAc-containing N-glycans on the common leukocyte antigen CD45 of B cells, leading to diminished intracellular calcium levels and ultimately inhibiting B cell activation [28]. This report was consistent with the reduction in cytosolic Ca^2+^ levels in our current study. Additionally, several studies revealed the distribution of IgM on B-cell surface membranes which form the nanoscale clusters and act as BCRs of primary B cells [29,51,52]. Using dual-color direct stochastic optical reconstruction microscopy, Cao and colleagues confirmed that Gal-9 can also directly bind to the IgM-BCR of murine B cells [29], nearly resulting in a complete abolishment of the BCR activation [29]. Due to the BCR-mediated Ca^2+^ influx as the critical signal for B cell activation [53], the immune modulation of CB-SCs on activated B cells primarily targets the regulation of intracellular Ca^2+^ levels through the Gal-9-mediated pathway, leading to dampened B-cell responses and shaping their differentiation. These novel molecular mechanisms will facilitate the clinical translation of Stem Cell Educator therapy to treat T1D and other autoimmune diseases.

## 4. Materials and Methods

### 4.1. B-Cell Isolation and Culture

Human peripheral blood mononuclear cells (PBMCs) (*N* = 12, aged from 31 to 64 years with an average of 46.83 ± 9.67 years old, male =7, female = 5) were isolated by Ficoll-hypaque density gradient (GE Healthcare, Chicago, IL, USA) from the buffy coats (Appendix A) purchased from the New York Blood Center (NYBC, New York, NY, USA). NYBC have received all accreditations for human blood collections and distributions and the methodology to clarify the sources of the cells, with institutional review board (IRB) approval and signed consent forms from donors. The isolations of PBMCs were performed as previously described [2]. PBMC cell suspensions were pre-treated with anti-CD19-conjugated microbeads (Miltenyi Biotec, Auburn, CA, USA) according to the manufacturer’s instructions. The purified CD19^+^ B cells were cultured in the chemically-defined and serum-free X-VIVO 15 medium (Lonza, Walkersville, MD, USA).

### 4.2. Proliferation Assay

To examine the effects of CB-SCs on B-cell proliferation, B cells were stimulated by the following combination at 37 °C and 5% CO_2_ conditions: the goat antihuman IgM F (ab’)2 (10 µg/mL), recombinant CD40L (rCD40L 1 μg/mL), IL-2 (10 ng/mL), IL-10 (20 ng/mL), and IL-21 (50 ng/mL) in the presence of the treatment with CB-SC for 7 days at the respective CB-SC/B cell ratios of 1:2, 1:5, and 1:10 in duplicates. The stimulated and unstimulated B cells in the absence of CB-SCs served as positive and negative controls, respectively. To detect the B-cell proliferation, the purified B cells were initially labeled with carboxyfluorescein succinimidyl ester (CFSE) (Life Technologies, Carlsbad, CA, USA), according to the manufacturer’s protocol. Consequently, the proliferation of B cells was detected by flow cytometry.

### 4.3. Culture of CB-SC and Coculture of CB-SCs with B Cells

The culture of CB-SCs was performed as previously described [2]. In brief, human umbilical cord blood units were collected from healthy donors and purchased from the Cryo-Cell international blood bank (Oldsmar, FL, USA). Cryo-cell has received all accreditations for cord blood collections and distributions, with hospital institutional review board approval and signed consent forms from donors. Mononuclear cells were isolated with Ficoll-hypaque (γ = 1.077, GE Health), and red blood cells were lysed using the ammonium-chloride-potassium (ACK) lysis buffer (Lonza, MD, USA). The remaining mononuclear cells were seeded in 150 × 15 mm style non-tissue culture-treated Petri dishes or non-tissue culture-treated 24-well plates (REF 351147, Corning Incorporated, Corning, NY, USA) at 1 × 10^6^ cells/mL. Cells were cultured in X-VIVO 15 chemically-defined serum-free culture medium and incubated at 37 °C with 8% CO_2_ for 10–14 days.

For the coculture of CB-SC with B cells: Initially, cord blood-derived mononuclear cells were planted in non-tissue culture-treated 24-well plates at 1 × 10^6^ cells per well and cultured in serum-free X-VIVO 15 media at 37 °C and 8% CO_2_ conditions for 10–14 days. Normally, CB-SCs reach above 80% confluence after culture for 10–14 days. The number of CB-SCs at 80% confluence was about 1 × 10^5^ cells/1 mL/well. All floating cells and debris were washed away, with the purified CB-SCs adhering to the bottom of plate and ready for the coculture with B cells. The starting cell concentration for B cells was 2 × 10^5^ B cells per well for a CB-SC/B cell ratio of 1:2; 5 × 10^5^ B cells per well for a CB-SC/B cell ratio of 1:5; and 10 × 10^5^ B cells per well for a CB-SC/B cell ratio of 1:10. There was no changing the media during the coculture for 7 days. CB-SCs tightly adhered to the surface of non-tissue culture plate and were resistant to the detachment with EDTA-trypsin, making it easy to separate CB-SCs from B cells for analysis in following experiments.

### 4.4. Quantitative Real-Time PCR Assay

The mRNA expressions of the galectin family were analyzed by quantitative real-time PCR as previously described [46]. Total RNA was extracted from CB-SCs using RNeasy mini Kit (Qiagen, Redwood City, CA, USA). The purity of RNA was routinely tested with a NanoDrop Spectrophotometer, with the ratio of absorbance at 260 nm and 280 nm normally at ~2.0. Then, 500 ng RNA was prepared for the cDNA transcription in 20 μL total volume. Consequently, the cDNA was diluted to 100 μL with DEPC water, and 1 μL of diluted cDNA was utilized for each qRT-PCR. First-strand cDNA were synthesized from total RNA using an iScript gDNA Clear cDNA synthesis Kit according to the manufacturer’s instructions (Bio-Rad, Hercules, CA, USA). Real-time PCR was performed using the StepOnePlus Real-time PCR system (Applied Biosystems, Waltham, MA, USA) under the following conditions: 95 °C for 10 min, then 40 cycles of 95 °C for 15 s, and 60 °C for 60 s. cDNA was amplified with the validated specific primers (Table 1) [54]. β-actin was used as controls.

### 4.5. Assay for Antibody Production

To detect the antibodies produced by B cells, B cells were stimulated by the following combination: goat antihuman IgM F (ab’)2 (10 µg/mL), recombinant CD40L (1 μg/mL), IL-2 (10 ng/mL), IL-10 (20 ng/mL), and IL-21 (50 ng/mL) in the presence or absence of the treatment with CB-SCs in 24-well plates, with 500 µL chemically-defined serum-free culture X-VIVO 15 medium (Lonza, Walkersville, MD, USA) per well. After the treatment for 7 days, the supernatants were collected to determine the levels of antibody productions (e.g., IgG1, IgG2, IgG3, IgG4, IgA, and IgM) by using LEGENDplexTM Human Immunoglobulin Isotyping Panel according to the manufacturer’s manual (Biolegend, San Diego, CA, USA). The Gallios Flow Cytometer was utilized to analyze the data according to the manufacturer’s recommended protocol. 

### 4.6. Blocking Experiments with Gal-9 Antibody

To determine whether Gal-9 contributes to the immune suppression of CB-SCs on the activated B cells, the purified CD19-positive B cells were activated with goat antihuman IgM F(ab’)2 (10 µg/mL), recombinant CD40L (rCD40L 1 μg/mL), IL-2 (10 ng/mL), IL-10 (20 ng/mL), and IL-21 (50 ng/mL) in the presence or absence of CB-SCs at a ratio of 1:2 in 24-well plate or 6-well plates, with or without adding Gal-9 mAb (10 µg/mL, Biolegend, San Diego, CA, USA). The blocking effects of Gal-9 mAb on the B-cell proliferation were examined by flow cytometry with CFSE staining.

To further explore the blocking effects of Gal-9 mAb, the activated B cells were characterized for their cytoplasmic and mitochondrial Ca^2+^ levels as well as mitochondrial membrane potential (Δψm) by using flow cytometry as previously described [2]. Briefly, after the 4 h treatment, B cells were stained with fluorescence dyes, including Fluo-4 (ThermoFisher Scientific, Waltham, MA, USA) for cytoplasmic Ca^2+^, Rhod-2 (ThermoFisher Scientific, Waltham, MA, USA) for mitochondrial Ca^2+^, and tetramethylrhodamine ethyl ester (TMRE) (Abcam, Waltham, MA, USA) for detection of mitochondrial membrane potential, respectively. For Fluo-4 and Rhod-2 staining, B cells were collected for staining with Fluo-4 or Rhod-2, respectively, in X-VIVO 15 media with a working concentration of 1 μM at 37 °C for 30 min. Then, the cells were washed with X-VIVO 15 medium and resuspended in 200 μL of medium for an additional 30 min at room temperature to enable the complete de-esterification of intracellular AM esters [2]. Finally, the cells were run for flow cytometry analysis.

### 4.7. Flow Cytometry

Phenotypic characterization of B-cell subsets was performed by flow cytometry [2] with specific markers including PE-conjugated mouse antihuman CD27 (Biolegend, San Diego, CA, USA) and APC-conjugated mouse antihuman IgD (Biolegend, San Diego, CA, USA). To determine the purity of CB-SCs, cells were examined by flow cytometry with CB-SC-associated markers, including PE-Cy7-conjugated mouse antihuman CD45 (Beckman Coulter, Brea, CA, USA), efluor660-conjugated rat antihuman OCT3/4 (ThermoFisher Scientific, Waltham, MA, USA), Alexa Fluor™ 488-conjugated rat antihuman SOX2 (ThermoFisher Scientific, Waltham, MA, USA), BV421-conjugated mouse antihuman CD34 (Biolegend, San Diego, CA, USA), PE-conjugated mouse antihuman CD270 (ThermoFisher Scientific, Waltham, MA, USA), and eFluor 450-conjugated mouse antihuman CD274 (ThermoFisher Scientific, Waltham, MA, USA) mAbs. Isotype-matched immunoglobulin (IgGs) served as controls.

### 4.8. Statistical Analysis

Statistical analysis of data was performed with the GraphPad Prism 8 (version 8.0.1) software. The normality test of samples was evaluated using the Shapiro–Wilk test. Statistical analysis of data was performed using the two-tailed paired Student’s *t*-test for the statistical analysis of two groups and one-way ANOVA test for multiple groups to determine statistical significance for parametric data between untreated and treated groups. The Mann–Whitney U test was utilized for non-parametric data. All experiments were performed with two technical replicates for each biological sample. The number “*N*” in each figure legend represented the biological replicates for each experiment. Values were given as mean ± SD (standard deviation). Statistical significance was defined as *p* < 0.05.

## 5. Conclusions

Stem Cell Educator therapy has been unutilized to treat T1D and multiple autoimmune- and inflammation-associated diseases [1], of which the pathogenesis involves T cells, B cells, and monocytes/macrophages. The current study revealed that CB-SCs directly displayed multiple immune modulations on B-cell proliferation and differentiation and antibody productions through the Gal-9-mediated mechanism and calcium flux/Akt/Erk1/2 signaling pathways. These findings lead to a better understanding of the molecular mechanisms of Stem Cell Educator therapy to treat T1D and other autoimmune diseases (e.g., myasthenia gravis and lupus) in clinics. Additionally, Stem Cell Educator therapy may fundamentally correct the activated B cell-mediated autoimmunity and reduce the auto-antibody productions, without the safety concerns that are associated with using steroid and conventional immunotherapies.

## Figures and Tables

**Figure 1 ijms-25-01830-f001:**
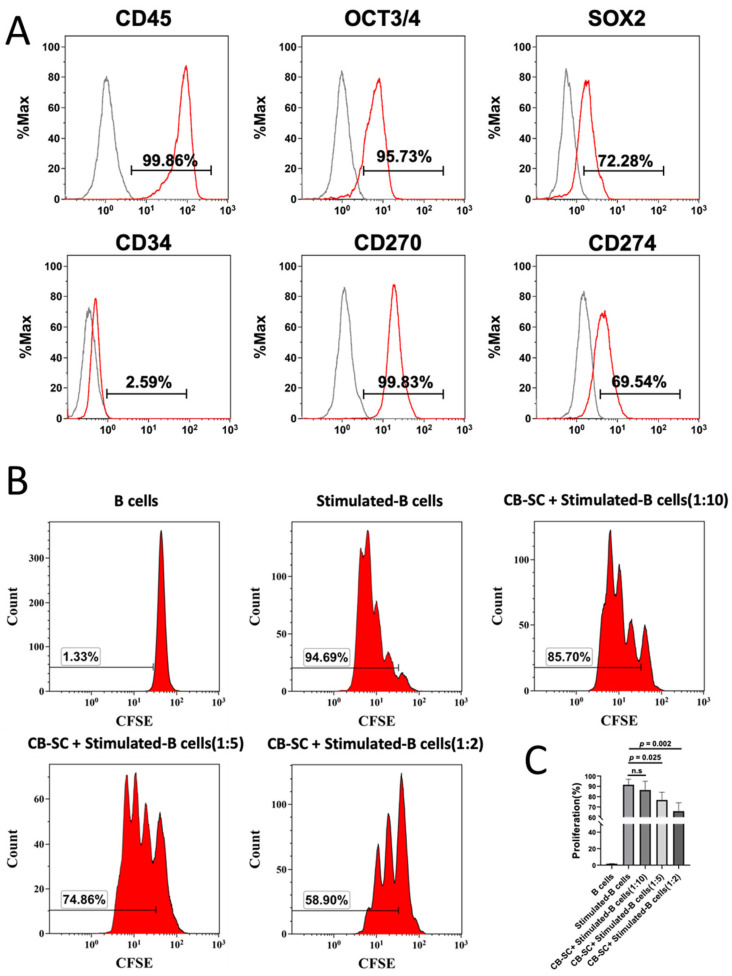
Immunosuppression of cord blood-derived stem cell on the B cell. (**A**) Phenotypic characterization of CB-SC with high purity. CB-SCs were analyzed by flow cytometry with associated markers, including common leukocyte antigen CD45, embryonic stem cell markers OCT3/4 and SOX2, hematopoietic stem cell marker CD34, and the immune modulation-related markers CD270 and CD274. Isotype-matched immunoglobulin G (IgGs) served as control. Data were represented from four experiments with similar results. (**B**) Suppression of B-cell proliferation by CB-SC. The carboxyfluorescein succinimidyl ester (CFSE)-labeled B cells were stimulated to proliferate with activation cocktails in the presence of different ratios of CB-SCs. Untreated B cells served as negative control. Histograms of flow cytometry were representative of five experiments with similar results. (**C**) Quantitative analysis of B-cell proliferation shows a remarkable decrease in B cell expansion after the treatment with CB-SCs at the different ratios of CB-SCs: B cells of 1:10 (n.s., *p* > 0.05, *N* = 3), 1:5 (*p* = 0.025, *N* = 3), and 1:2 (*p* = 0.002, *N* = 3).

**Figure 2 ijms-25-01830-f002:**
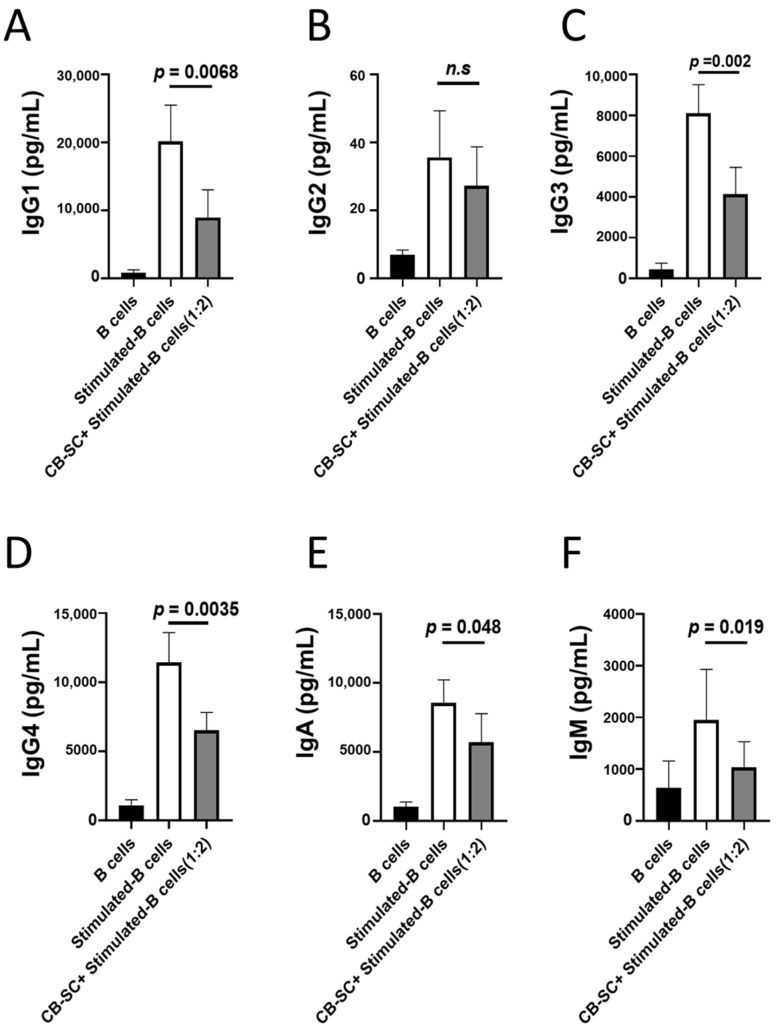
Inhibition of B-cell immunoglobulin production by CB-SCs. B cells were stimulated in the presence of cocktails (anti-IgM, rCD40L, IL-2, IL-4, IL-10, IL-21). CB-SCs markedly inhibit Ig production of stimulated B cells at a ratio of CB-SC/B-cells of 1:5. Untreated B cells served as negative control. (**A**) CB-SCs inhibit IgG1 production. *p* = 0.0068 (*N* = 4). (**B**) CB-SCs inhibit IgG2 production. *p* = 0.330 (N = 6). (**C**) CB-SCs inhibit IgG3 production. *p* = 0.002 (*N* = 4). (**D**) CB-SCs inhibit IgG4 production. *p* = 0.0035 (*N* = 6). (**E**) CB-SCs inhibit IgA production. *p* = 0.048 (*N* = 4). (**F**) CB-SCs inhibit IgM production. *p* = 0.019 (*N* = 6). n.s., *p* > 0.05.

**Figure 3 ijms-25-01830-f003:**
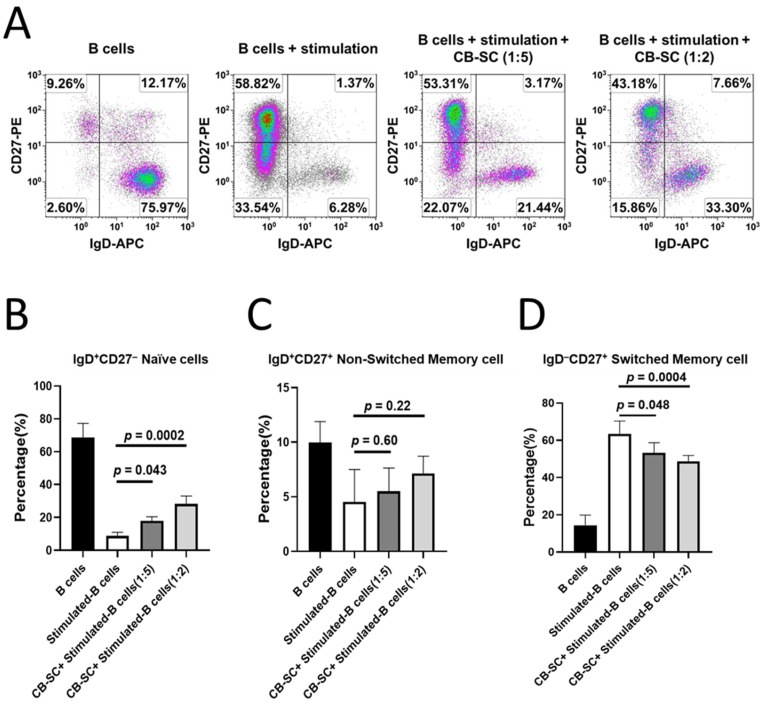
Modulation of different B cell subpopulations by CB-SCs. (**A**–**D**) B cells were collected for flow cytometry after the coculture with CB-SCs for 7 days. Only the propidium iodide (PI)-negative viable cells were gated for flow cytometry analysis. (**A**) Upregulation of the percentage of naïve B cells by CB-SCs and downregulation of the percentage of switched B cells. Histograms of flow cytometry were representative of four experiments with similar results. Isotype-matched IgGs served as negative controls. *N* = 4. (**B**) Increase in the percentage of naïve B cells after the treatment with CB-SCs at a ratio of 1:5 (*p* = 0.043, *N* = 4), 1:2 (*p* = 0.0002, *N* = 4). (**C**) There is no significant effect on the percentage of non-switched B cells. *N* = 4. (**D**) Decrease in the percentage of switched B cells (*N* = 4). Results are given as mean ± SD. *P* < 0.05 as a significant difference.

**Figure 4 ijms-25-01830-f004:**
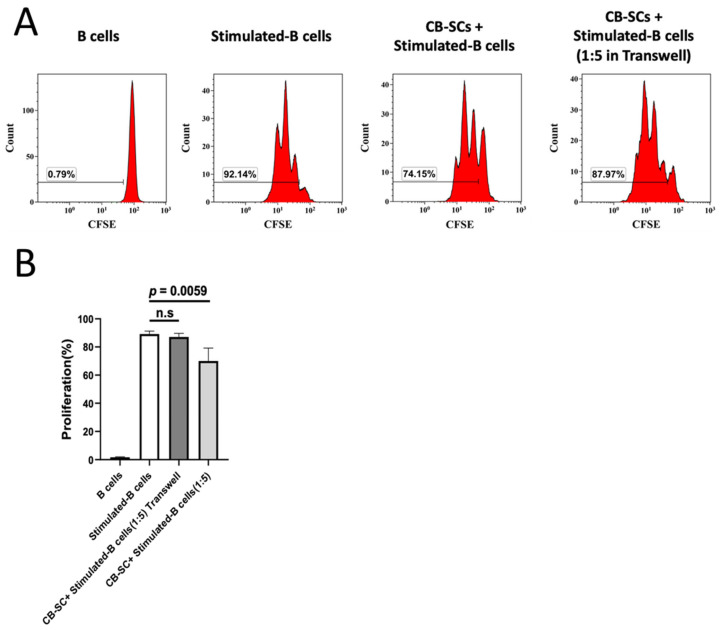
Cell–cell-contact-mediated mechanism contributes to the CB-SC-mediated immune suppression. (**A**) CB-SCs cocultured with CFSE-labeled stimulated B cells in transwells. CB-SCs directly cocultured with CFSE-labeled stimulated B cells served as control. Both unstimulated and stimulated B cells without CB-SC coculture served as additional control. Histograms of flow cytometry were representative of four experiments with similar results. (**B**) CB-SCs failed to suppress the proliferation of stimulated B cells in transwell coculture system at a ratio of CB-SC/B cells of 1:5. In contrast, the direct coculture of CB-SCs with stimulated B cells displayed the marked inhibition of B-cell proliferation at the same ratio (*p* = 0.0059, *N* = 4), n.s., *p* > 0.05.

**Figure 5 ijms-25-01830-f005:**
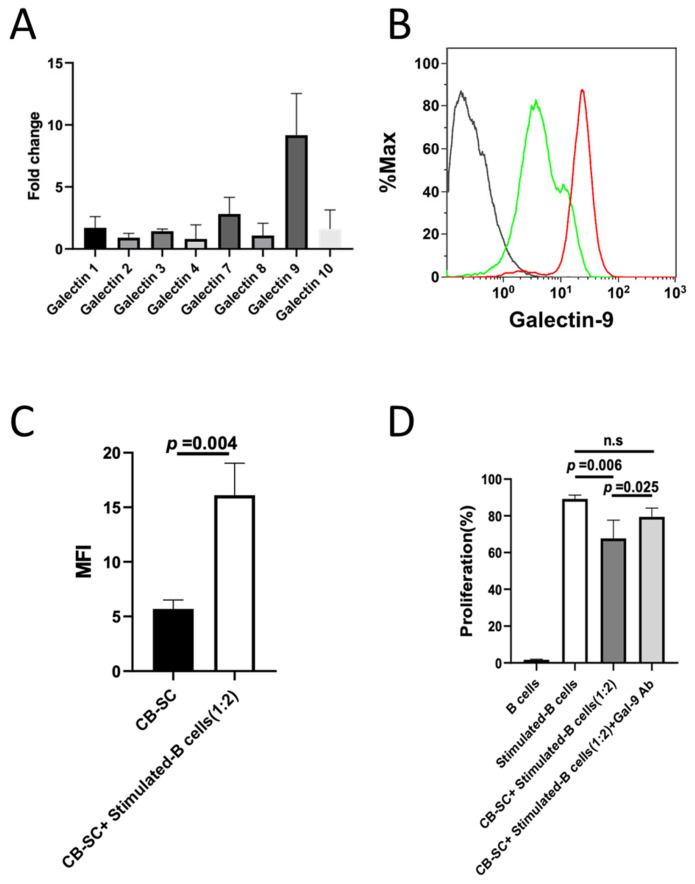
Gal-9 expressed on CB-SC acts as the key mediator for the CB-SC-induced B-cell suppression. (**A**) Changes in the levels of galectin expressions after coculture of CB-SCs with the stimulated B cells for 2 days (*N* = 3). (**B**) Upregulated level of Gal-9 expression on CB-SCs after coculture with stimulated B cells (red line) relative to that on the untreated CB-SC (green line). Isotype-matched IgG served as negative control. *N* = 3. (**C**) Increased level of Gal-9 expression (median fluorescence intensity, MFI) on CB-SCs after coculture with stimulated B cells. The data are given as mean ± SD of three experiments with CB-SC (*N* = 3)-treated B cells (*N* = 3). (**D**) Stimulated B cells were cocultured with CB-SCs at ratio of 1:2 in the presence or absence of Gal-9 mAb blocking. The inhibition of CB-SCs on B-cell proliferation was abolished after blocking with Gal-9 mAb (stimulated B cells with CB-SC *vs*. stimulated B cells + CB-SC + Gal-9 mAb, *p* = 0.025, *N* = 4), n.s., *p* > 0.05. The carboxyfluorescein succinimidyl ester (CFSE)-labeled B cells were stimulated to proliferate with activation cocktails in the presence of CB-SCs at ratio of 1:2 in 24-well plate at 37 °C with 8% CO_2_ for 7 days and followed by flow cytometry. Untreated B cells served as negative control.

**Figure 6 ijms-25-01830-f006:**
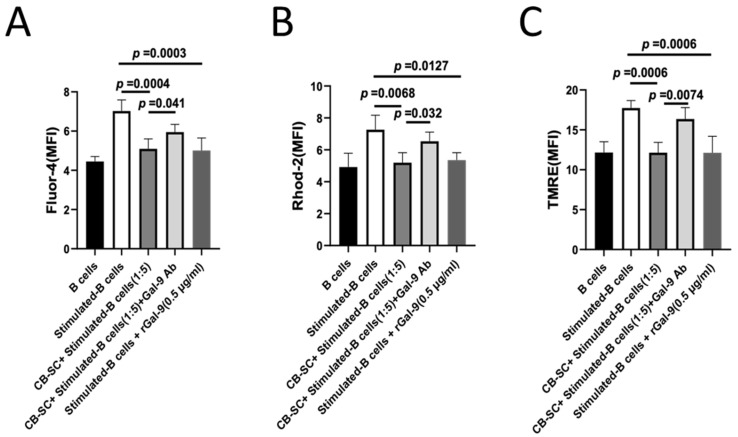
Gal-9 expressed on CB-SCs contributes to the modulation of Ca^2+^-associated signaling pathways in stimulated B cells. (**A**) Flow cytometry analysis of cytoplasmic Ca^2+^ with Fluor-4 staining shows the remarkable decrease in the median fluorescence intensity (MFI) value of Fluor-4^+^ B cells after the treatment with CB-SCs (*p* = 0.0004, *N* = 3) or 0.5 μg/mL rGal-9 (*p* = 0.0003, *N* = 3), but they are markedly increased after blocking with Gal-9 mAb. (**B**) Flow cytometry analysis of mitochondrial Ca^2+^ with Rhod-2 staining shows the substantial decline in the MFI value of Rhod-2^+^ B cells after the treatment with CB-SCs (*p* = 0.0068, *N* = 3) or 0.5 μg/mL rGal-9 (*p* = 0.0127, *N* = 3), but they are clearly improved after blocking with Gal-9 mAb. (**C**) Downregulation of the mitochondrial membrane potential in the stimulated B cells after the treatment with CB-SCs, which was performed in the Gal-9-dependent manner as demonstrated by flow cytometry analysis after staining with TMRE (*N* = 3).

**Table 1 ijms-25-01830-t001:** List of primer sequences for RT-PCR analysis of human galectins.

	Forward Primer (5′–3′)	Reverse Primer (5′–3′)
Galectin 1	TGCAACAGCAAGGACGGC	CACCTCTGCAACACTTCCA
Galectin 2	GATGGCACTGATGGCTTTG	AGACAATGGTGGATTCGCT
Galectin 3	CAGAATTGCTTTAGATTTCCAA	TTATCCAGCTTTGTATTGCAA
Galectin 4	CGAGGAGAAGAAGATCACCC	CTCTGGAAGGCCGAGAGG
Galectin 7	CAGCAAGGAGCAAGGCTC	AAGTGGTGGTACTGGGCG
Galectin 8	CTTAGGCTGCCATTCGCT	AAGCTTTTGGCATTTGCA
Galectin 9	CTTTCATCACCACCATTCTG	ATGTGGAACCTCTGAGCACTG
Galectin 10	AGTGTGCTTTGGTCGTCGT	ATGCTCAGTTCAAATTCTTGG
Galectin 12	TGTGAGCCTGAGGGACCA	GCTGAGATCAGTTTCTTCTGC
Galectin 13	CTTTACCCGTGCCATACAA	GTGGGTCATTGATAAAAGAGTG
Galectin 14	CCTTGATGATTGTGGTACCAT	GTGGGTCCTTGACAAAAGTG
Beta actin	CATGTACGTTGCTATCCAGGC	CTCCTTAATGTCACGCACGAT

## Data Availability

The data that support the findings of this study are available from the corresponding author upon request.

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
