# Peer review of "Suppression of B-Cell Activation by Human Cord Blood-Derived Stem Cells (CB-SCs) through the Galectin-9-Dependent Mechanism"

_ijms, 2024, doi:10.3390/ijms25031830_

Round 1
Reviewer 1 Report
Comments and Suggestions for Authors
The paper from Hu and colleagues analyzed the CB-SC immunomodulatory effect on B-cells. The work is scientifically sound and fills the gap in the undertanding of Stem Cell Educator therapy action on immune cells opening the way for clinical use in treating autoimmune disease. Even if, in my opinion, the molecular mechanisms through which CB-SCs exert their immunomodulation effect need to be further explored, I think that the present article deserves the publication.
Only some concerns to be addressed:
1. In the introduction it is not clear the reason why the authors focused on galectins. Please, better explain the link between B cells and galectins signalling pathway.
2. There is no any ethical committee reference for buffy coat use for B cell isolation.
3. paragraph 2.2: Whe the proliferation assay has been performed in duplicates? Three technical replicates are preferred.
4. paragraph 2.3: It is not clear how the authors selected the CB-SC from mononuclear cells? Please clarify.
5. Fig. 1 A and B; Fig 3 A: Fig. 4A: For the flow cytometry experiments, it is ok to use a representative graph, but it is preferred to report the percentages as mean+-SD/SEM of the 4/5 experiments.
6. Line 373: The authors wrote "As shown in Figure 6D” but no panel D is present in figure 6.
7. In Fig S2 the "n" of the experiment is missing.
Author Response
Many thanks for your kind consideration and comments that were very helpful for us to improve the quality of this manuscript. Please see our point-by-point responses in attachment.
Yong

Reviewer 2 Report
Comments and Suggestions for Authors
This manuscript explores the effects of cord blood-derived stem cells on human B cells, with a primary focus on the role of galectin-9 in this process. The major observation is that the suppression of B-cell proliferation when cocultured with CB-SC. This interaction leads to a notable downregulation in immunoglobulin production from activated B cells. Moreover, the treatment with CB-SC induces phenotypic changes in B cells, marked by an increased percentage of IgD+CD27- naïve B cells and a decreased percentage of IgD-CD27+ switched B cells. Following this observation, the authors studied the mechanism behind this immune suppression and pinpointed the central role of galectin-9 in this process. As evidenced by blocking experiments using anti-galectin-9 monoclonal antibody, the immune-modulating effect of CB-SC on B cells is largely dependent on galectin-9. The study also uncovered alterations in calcium levels and signaling pathways in B cells treated with CB-SC, alongside a decline in mitochondrial membrane potential and decreased levels of phosphorylated Akt and Erk1/2 signaling proteins. These findings are new and have substantial clinical implications. Overall, this is a very interesting manuscript to be further considered. I have a few suggestions below to further improve the quality and readability of this work.
1. The authors must clearly state the rationale behind their hypotheses, such as why they prioritize galectins over other molecules. The current introductory paragraph (beginning at line 302) appears somewhat odd. Numerous molecules have been identified as regulators of immune tolerance and homeostasis. It makes no sense to target galectin only. Therefore, the authors should better articulate their rationales and provide relevant data or early observations if necessary.
2. The authors should more comprehensively discuss the limitations of their study. The current discussion section fails to address this aspect adequately.
3. The authors should expand the content in the conclusions, for instance, by including potential opportunities for future development.
Author Response

(The authors gave the same response as above.)

Reviewer 3 Report
Comments and Suggestions for Authors
The main question of galectin involvement to mechanisms MSC- mediated of B cell immunomodulation were raised and resolved.
The work has the description of all standard part of original article.
The description of galectin MSCs from cord blood and there effects on B-cell function is rather new. Earlier there were descriptions of the same effects on other models predominantly in vivo.
The article stress the possibilities for galectins to participate in MSC -mediated immunosuppression upon B cell function in vitro.
Metodology in article as described is adequated to aid and tasks of study.
Conclusions and references are appropriated.
Author Response

(The authors gave the same response as above.)
